# Action-Dependent Optimality-Preserving Reward Shaping

**Grant C. Forbes** [1]   **Jianxun Wang** [1]   **Leonardo Villalobos-Arias** [1]   **Arnav Jhala** [1]   **David L. Roberts** [1]

## Abstract

Recent RL research has utilized reward shaping–particularly complex shaping rewards such as intrinsic motivation (IM)–to encourage agent exploration in sparse-reward environments. While often effective, "reward hacking" can lead to the shaping reward being optimized at the expense of the extrinsic reward, resulting in a suboptimal policy. Potential-Based Reward Shaping (PBRS) techniques such as Generalized Reward Matching (GRM) and Policy-Invariant Explicit Shaping (PIES) have mitigated this. These methods allow for implementing IM without altering optimal policies. In this work we show that they are effectively unsuitable for complex, exploration-heavy environments with long-duration episodes. To remedy this, we introduce Action-Dependent Optimality Preserving Shaping (ADOPS), a method of converting intrinsic rewards to an optimality-preserving form that allows agents to utilize IM more effectively in the extremely sparse environment of Montezuma's Revenge. We also prove ADOPS accommodates reward shaping functions that cannot be written in a potential-based form: while PBRS-based methods require the cumulative discounted intrinsic return be independent of actions, ADOPS allows for intrinsic cumulative returns to be dependent on agents' actions while still preserving the optimal policy set. We show how action-dependence enables ADOPS's to preserve optimality while learning in complex, sparse-reward environments where other methods struggle.

[1]Department of Computer Science, North Carolina State University, Raleigh, USA. Correspondence to: Grant C. Forbes <gforbes@ncsu.edu>, Jianxun Wang <jwang75@ncsu.edu>, Leonardo Villalobos-Arias <lvillal@ncsu.edu>, Arnav Jhala <ahjhala@ncsu.edu>, David L. Roberts <dlrober4@ncsu.edu>.

*Proceedings of the $42^{nd}$ International Conference on Machine Learning*, Vancouver, Canada. PMLR 267, 2025. Copyright 2025 by the author(s).

## 1. Introduction

There is growing interest in the Reinforcement Learning (RL) literature in using reward shaping to train agents in sparse-rewards environments that would otherwise be intractable (Mataric, 1994; Randløv & Alstrøm, 1998; Bellemare et al., 2016); specifically, interest in Intrinsic Motivation (IM): complex, non-Markovian reward functions used to encourage exploration in sparse reward environments (Pathak et al., 2017; Burda et al., 2018).

It has been noted that both traditional reward shaping (Randløv & Alstrøm, 1998) and IM (Burda et al., 2019) can be "hacked," with the agent learning to optimize the shaping reward at the expense of the actual reward. Potential-Based Reward Shaping (PBRS) aims to rectify this and provide a form for reward-shaping terms in which they can be guaranteed to not alter the set of optimal policies of the underlying environment. To apply PBRS to IM, previous work on Potential-Based Intrinsic Motivation (PBIM) (Forbes et al., 2024a) and Generalized Reward Matching (GRM) (Forbes et al., 2024b) implement arbitrary intrinsic rewards while preserving the theoretical optimality guarantees of PBRS. These methods have proven effective at speeding up training and preventing divergence from an optimal policy in grid world and cliff walking domains, with a tabular exploration reward and Random Network Distillation (RND), respectively. However, their efficacy is still untested in more complex environments. Policy-Invariant Explicit Shaping (PIES) (Behboudian et al., 2022), another method developed for implementing shaping rewards without altering the optimal policy, has similarly been tested thus far only in relatively simple environments, and not in any environments wherein training with intrinsic reward is itself essential in consistently obtaining any extrinsic rewards in agent-environment interactions during training. Testing in Montezuma's Revenge, a benchmark environment that has been widely acknowledged (Mnih et al., 2015; Burda et al., 2019) to possess these characteristics, we find that all of these methods, while preserving the optimal policy set, detract from the agent's ability to learn to the extent that none of them can outperform an agent training on RND alone.

Motivated by this, we develop Action-Dependent Optimality-Preserving Shaping (ADOPS), a method for converting any arbitrary shaping reward (including IM)

to a form that preserves optimality. ADOPS functions by consulting an agent's critic networks' estimations of the extrinsic and intrinsic value functions, and using these estimations, actively adjusting the intrinsic reward seen by the agent if and only if that reward would cause an action to be preferred when it would not be preferred by external rewards alone, or vice-versa. ADOPS improves on existing methods in several key ways:

1. **We forego the need for several key assumptions required for previous methods to ensure optimality.** These include the requirement that the environment be episodic and that the underlying IM is "future-agnostic."

2. **We encompass a provably wider set of optimality-preserving shaping functions than prior methods.** While GRM and PBIM both ensure and require that the expected intrinsic return at any given time step is action-independent, our method produces shaping functions whose returns can be action-dependent, yet still preserve optimality. Additionally, while PIES preserves optimality by eventually returning no shaping rewards at all, our method allows the agent to receive shaping rewards for an arbitrarily long amount of time. We argue that both of these improvements over prior methods are key to ADOPS's ability to outperform the baseline in complex, extremely sparse environments.

3. **ADOPS empirically improves performance over baseline IM in complex, extremely sparse environments where preexisting methods for preserving optimality fail.** We empirically demonstrate improvement over other optimality-preserving methods and the baseline IM in Montezuma's Revenge, and thus achieve a new SOTA in implementing IM while preserving the optimal policy.

## 2. Background and Related Work

Here, we briefly discuss the relevant background literature more broadly. In Section 3, we describe more closely the three specific methods we are building off of, and which we test against ADOPS in Section 6.

### 2.1. Intrinsic Motivation

Many problems in reinforcement learning are reward-sparse (Mataric, 1994) and are difficult for agents to properly learn. Sparse rewards can lead to an agent never learning a viable policy. One method to remedy this is by granting a shaping reward: an extra reward, added to the native environment reward, to decrease sparseness, and convey more useful information.

Intrinsic Motivation is a subfield of reward shaping that aims to navigate sparse-reward environments by giving the agent additional "intrinsic" reward shaping functions. These are usually distinguished from traditional reward shaping func-

tions both by their generality, as most of them are domain-agnostic, and by their complexity. IM methods are almost universally non-Markovian, often involving the agent's entire training history, and involving additional neural networks and/or auxilliary learning algorithms.

Intrinsic motivation is often used as a means to reward exploration of the environment. IM methods such as count-based exploration (Bellemare et al., 2016), Intrinsic Curiosity Module (ICM) (Pathak et al., 2017), and Random Network Distillation (RND) (Burda et al., 2018; 2019) incentivize visiting unfamiliar states. More recent IM works implement increasingly complex strategies (Badia et al., 2020; Yuan et al., 2023).

While useful for exploring, IM algorithms, including RND, are known to alter the optimal policy of the underlying environment. Take for instance the "noisy TV problem" (Burda et al., 2018), in which an agent prioritizes an inrinsically-rewarding stochastic area of the state space at the expense of pursuing external rewards. A version of this problem exists in Montezuma's Revenge with RND in particular, deemed "dancing with skulls," in which the agent tends to actively pursue "risky" behavior (Burda et al., 2019).

There have been some efforts to mitigate the noisy TV problem: Chen et al. (Chen et al., 2022) introduce EIPO, which automatically adjusts the scaling factor of the IM, reducing the reward when exploration is unnecessary and increasing it when the agent must explore, and Le et al. (Le et al., 2024) mitigate it using "surprise novelty." However, neither of these approaches provide theoretical guarantees that the optimal policy set is unchanged, and they both require non-trivial additional (neural network) architecture and computational overhead. Forbes et al. (Forbes et al., 2024a) introduce PBIM, a Potential-Based-Reward-Shaping (PBRS)-based approach that overcomes both of these limitations, and further expand it to the more general method of Generalized Reward Matching (GRM) (Forbes et al., 2024b). However, these works don't demonstrate efficacy at scale. We discuss these works in further detail in Section 3.

### 2.2. Potential-Based Reward Shaping

Potential-Based Reward Shaping is a subfield of reward shaping that specifically studies reward-shaping terms that are theoretically guaranteed to maintain the optimal policy set of the underlying environment. The foundational paper in the field (Ng et al., 1999) explores how adding a reward shaping term to an MDP can alter the optimal policy. The study uncovers that the optimal policy will remain unchanged in infinite-horizon MDPs, or those with an absorbing state, if the shaping reward follows the form of a state-based potential $F = \gamma\Phi(s') - \Phi(s)$. This is further extended by Wiewiora et al (Wiewiora, 2003) by creating a state-action potential $\Phi(s, a)$, whereas the original

was state-only. Such potentials, however, must be removed from the final Q-values learned by the model to guarantee policy invariance. Similarly, Devlin et al. (Devlin & Kudenko, 2012) extend the original potential notation by adding time-dependence, $\Phi(s, t)$, similarly guaranteeing policy invariance. Harutyunyan et al. (Harutyunyan et al., 2015) uncovered a method to convert any Markovian reward function into a potential-based version that follows the form $\Phi(s, a, t)$. A later study by Behboudian et al. (Behboudian et al., 2022) however demonstrates that this potential-based conversion can, in fact, alter the optimal policy. In this same article, the authors propose Policy-Invariant Explicit Shaping (PIES), an algorithm that decrements the magnitude of shaping rewards to guarantee policy invariance by the end of training. We discuss this work more closely in Section 3.

PBRS was extended to the finite-horizon setting in (Grzes, 2017), and this finite-horizon setting was then later shown to be able to accomodate most IM terms' complex variable-dependence via PBIM (Forbes & Roberts, 2024; Forbes et al., 2024a) and later GRM (Forbes et al., 2024b).

In the next section, we highlight three of the most recent potential-based approaches: PIES, PBIM, and GRM. These are all "plug-and-play" approaches, signifying that they are generally adaptable methods for converting a reward shaping function (or IM) into a form that preserves the optimal policy (provided some assumptions are met). They will be the baseline algorithms that we compare our method against.

## 3. Preliminaries and Previous Methods

We define a Markov Decision Process (MDP) as a tuple $M = (S, S_0, A, P, \gamma, R)$, where $S$ is the state space, $S_0 \subseteq S$ is the set of start states, $A$ is the action space, $P(s_{t+1} = s'|s_t = s, a_t = a)$ is the probability of transitioning to state $s'$ by taking action $a$ in state $s$, $\gamma \in [0, 1]$ is the discount factor, and and $R$ is the reward function. An agent in an MDP follows a policy $\pi(a|s) : S \times A \to [0, 1]$, which gives the probability of taking action $a$ in state $s$. An optimal policy maximizes the value function:

$$V_M^\pi = \mathop{\mathbb{E}}_{a\sim\pi, s_0\sim S_0, s\sim P} \sum_{t=0}^{N-1} \gamma^t R_t, \tag{1}$$

where $R_t$ is the reward at time $t$, and $N$ is the number of time steps in a given episode. For infinite-horizon cases, the sum to $N - 1$ becomes an infinite sum.

Reward shaping methods, such as IM, add a reward $F_t$ on top of the regular MDP reward $R_t$. This defines a new MDP $M' = (S, S_0, A, P, \gamma, R')$, where

$$R_t' = R_t + F_t. \tag{2}$$

Thus an optimal policy in $M'$ may be suboptimal in $M$, and we are interested in constructing an $F_t$ such that this is provably not the case.

### 3.1. Potential-Based Intrinsic Motivation (PBIM)

Potential-Based Intrinsic Motivation (Forbes et al., 2024a), inspired by (Ng et al., 1999), proves that the optimal policy set of an environment will remain unchanged by the addition of a shaping reward in the form

$$F_t = \gamma\Phi_{t+1} - \Phi_t, \tag{3}$$

where $\Phi_t$ is a potential function that meets the boundary condition:

$$\mathop{\mathbb{E}}_{a\sim\pi, s\sim T, R_n\sim R} \left(\gamma^{N-t}\Phi_N - \Phi_t\right) = \Phi_t', \forall t \in (0, ...N-1), \tag{4}$$

and $\Phi_t'$ is an arbitrary function that is constant with respect to action $a_t$.

Any arbitrary shaping reward $F_t$ can be transformed into a potential-based form if the value of $F_t$ does not depend on any future actions that the agent takes (For example, in episodic environments where the shaping reward is given retroactively). That is, $F_t$ must meet

$$F_t \text{ is constant w.r.t. } a_{t'>t}, \forall t, t' \in (0, \ldots, N-1), \tag{5}$$

The authors of PBIM propose two transformation methods, normalized and non-normalized PBIM. Normalized PBIM is defined as

$$F_t'^{PBIM} = \begin{cases} \sum_{i=0}^{N-2} -\gamma^{i-N} F_i', & \text{if } t = N-1 \\ F_t - \bar{F}, & \text{if } t \neq N-1, \end{cases} \tag{6}$$

where $\bar{F}$ is an estimated average shaping reward for the current policy. $F_t'$ for non-normalized PBIM is defined identically, except that the $\bar{F}$ term is set to zero. They prove that using either of these conversion methods with an $F_t$ that meets the assumption in Equation 5 will result in a $F_t'$ that leaves the optimal policy unchanged if used as the shaping reward in Equation 2. Intuitively, both of these methods function by waiting until the last time step of an episode, then subtracting all the accumulated intrinsic rewards from the agent.

### 3.2. Generalized Reward Matching (GRM)

Generalized Reward Matching (Forbes et al., 2024b) is a broader optimality-preserving reward shaping method that encompasses both PBIM and all other PBRS-based methods. It defines a matching function $m_{t,t'} : N \times N \to [0, 1]$ that 'matches' every instance of the shaping reward at timestep $t'$ with an appropriately-discounted negative reward in a future

timestep $t$. The general form of a GRM shaping reward is

$$F_t^{'\text{GRM}} = F_t - \sum_{i=0}^{t} \gamma^{i-t} F_i m_{t,i}. \qquad (7)$$

GRM rewards are proven not to alter the optimal policy under two conditions:

1. Fully-matching: Each given shaped reward must be subtracted exactly once (Although it may be partially subtracted across various timesteps): $\forall_{t'} \sum_{j=t'}^{N-1} m_{j,t'} = 1$.

2. Future-agnostic: Each shaped reward can only be discounted in future timesteps: $\forall_{t,t'>t} \quad m_{t,t'} = 0$.

A family of GRM functions is introduced in the same study, where the shaping reward is 'matched' either after a delay of $D \geq 0$ timesteps, or at the end of an episode, whichever comes first. The corresponding shaping reward is defined as:

$$F_t^{'\text{GRM}}(D) = \begin{cases} F_t & \text{if } t < D \\ F_t - \gamma^{-D} F_{t-D} & \text{if } D \leq t < N-1 \\ \sum_{i=0}^{D-1} -\gamma^{i-D} F_{N-1-D+i} & \text{if } t = N-1. \end{cases} \qquad (8)$$

Note here, as noted in (Forbes et al., 2024b), that if $D \geq N-1$ for any given episode, then Equation (8) becomes equivalent to PBIM, while if $D = 0$, it becomes equivalent to having no shaping reward at all. In our experiments, we use this same parameterization as a representative sample of GRM methods.

### 3.3. Policy-Invariant Explicit Shaping (PIES)

Policy-Invariant Explicit Shaping (Behboudian et al., 2022) proposes learning from a shaping reward without altering the optimal policy by multiplying the expected return of that shaping reward by a scaling coefficient $\zeta$ that scales linearly to zero before the end of training. During training, PIES begins with $\zeta_0 = 1$, then for each episode $n$ during training, $\zeta$ is updated according to

$$\zeta_n = \begin{cases} \zeta_{n-1} - \frac{1}{C} & \text{if } \zeta_{n-1} > \frac{1}{C} \\ 0 & \text{otherwise.} \end{cases} \qquad (9)$$

Here, $C$ is a coefficient that controls how quickly $\zeta$ decays. Ideally, it will reach zero with enough training time left that the agent can converge to an optimal policy, if the reward shaping term is one that the agent would otherwise hack.

PBIM and GRM.[1] However, though it is a sufficient condition for the preservation of optimality, it is not a necessary condition, and we prove in Theorem B.1 that there will exist optimality-preserving reward-shaping terms that cannot be written as a difference of potentials that follow Equation (4) in any non-trivial environment.

Similarly, PIES preserves optimality by using only extrinsic reward in the latter stages of training: this is also clearly a sufficient condition to preserve optimality, but just as clearly not a necessary one. As such, it follows that many optimality-preserving reward-shaping functions cannot, in principle, ever be obtained from either PIES or the PBRS-based methods described in prior literature.

In this section, we examine the more general condition from which Equation 4 is derived, and derive a more general condition, which captures optimality-preserving reward-shaping potentials that are "action-dependent," and thus provably cannot be accommodated by prior plug-and-play optimality-preserving methods. In Section 5 we then introduce ADOPS, which can accommodate this more general form.

### 4.1. Conditions For Optimality-Preserving Reward Shaping Functions Beyond PBRS

We define the Q-function

$$Q_M^\pi(s, a, t) = R_M(s, a, t) + V_M^\pi(s', t+1) \qquad (10)$$

to be the expected discounted return of taking action $a$ in state $s$ of MDP $M$, then following policy $\pi$.[2] For simplicity of notation, we also define $Q_M^\pi(s_t, a_t, t) = Q_{M,t}^\pi = Q_M^\pi$. We notate the Q-function of taking an action and then following an optimal policy as $Q_M^{\pi^*} = Q_M^*$. We can then write the general condition for preserving optimality: we want to ensure the optimal policy set is the same for both the original MDP $M$ and the shaped MDP $M'$, that is:

$$\operatorname*{argmax}_a Q_M^* = \operatorname*{argmax}_a Q_{M'}^* \quad \forall s, t. \qquad (11)$$

This is the condition that all prior work in PBRS seeks to preserve. Note crucially that we are defining $\pi^*$ such that it is an optimal policy in the *original* environment $M$, not necessarily in an environment $M'$ with the shaping reward present. For clarity on this point, we define $Q_E^\pi = Q_M^\pi$ to be the extrinsic Q function, and $Q_{IE}^\pi = Q_{M'}^\pi = Q_E^\pi + Q_I^\pi$ to be the combined extrinsic and intrinsic Q functions. We define $V_{IE}^\pi, V_E^\pi$, and $V_I^\pi$ likewise. Thus, we have

$$\operatorname*{argmax}_a Q_E^* = \operatorname*{argmax}_a (Q_E^* + Q_I^*) \quad \forall s, t, \pi^*. \qquad (12)$$

## 4. More General Conditions For Optimality

A PBRS term that follows Equation (4) guarantees that the optimal policy set of the original MDP will remain unchanged. This includes the PBRS terms utilized in both

---

[1]The $\Phi_t$ terms in each of these methods are implicit, rather than explicit, but both methods are designed around ensuring that Equation 4 always holds.

[2]Note the direct $t$-dependence, as we're generally dealing with non-Markovian reward functions.

Note the enumeration over all $\pi^*$: this is because, while every optimal policy by definition has identical $Q_E^*$ and $V_E^*$ to every other optimal policy, they may differ from each other intrinsically, resulting in different $Q_I^*$ and $V_I^*$.

If we now define $\bar{a}$ as any action not in $\mathrm{argmax}_a \, Q_E^*$, then Equation 12 becomes equivalent to

$$V_{IE}^{\pi_1^*}(s,t) = V_{IE}^{\pi_2^*}(s,t) \quad \forall s, t, \pi_1^*, \pi_2^* \tag{13}$$
$$Q_{IE}^*(s, \bar{a}, t) < V_{IE}^*(s,t) \quad \forall s, \bar{a}, t, \pi^*. \tag{14}$$

Intuitively, the first of these conditions says that every action that would be optimal without IM must remain optimal after the addition of the IM, while the second condition says that any suboptimal action must remain suboptimal with the addition of any shaping reward.[3]

In prior work in PBRS, the optimal policy set is provably unchanged due to mathematical guarantees that $Q_{M'}^*$ differs from $Q_M^*$ by some term that is *independent* of the agent's actions at a given time step: see for example the $\Phi(s)$ of (Ng et al., 1999), or the $\Phi_t'$ of (Forbes et al., 2024a). In other words, all prior work in this area has met the condition of Equation 12 by removing $Q_I^*(a)$'s $a$-dependence, and essentially setting $Q_I^*(a) = V_I^* \quad \forall a$. While this is a sufficient condition to ensure that optimality is preserved (as this term then drops out of the $\mathrm{argmax}_a$, it is not a necessary one. As we will see, it leaves out a theoretically interesting and empirically useful subset of optimality-preserving reward shaping functions: those whose cumulative intrinsic returns are allowed to depend on the agent's actions. In Appendix B.1, we offer a formal proof that there exist optimality-preserving reward shaping functions that meet these conditions we have just provided, but which cannot be accounted for by any PBRS function. In the next section, we derive a new plug-and-play method of utilizing arbitrary IM functions while theoretically guaranteeing preservation of optimality, one that both overcomes the limitations of flexibility highlighted in Appendix B.1 and foregoes the need for several key assumptions (an episodic environment, and future-agnosticity as defined in Equation 5). We then show that our method speeds training efficiency over a baseline in a complex, difficult environment where other optimality-preserving methods fail.

# 5. ADOPS For Action-Dependent Optimality-Preserving Shaping

Here, we introduce our main contribution, and prove its efficacy at preserving the optimal policy.

---

[3]Implicit in the step from Equation 12 to Equation 13 is the fact that $Q_E^*(a^*) = V_E^* \quad \forall a^*, \pi^*$.

## 5.1. The 'Ideal' ADOPS Function

Inspired by the conditions in Equations 13 and 14, we first introduce an "ideal" reward-shaping conversion method that actively checks whether these conditions are satisfied, and if not, modifies the initial shaping reward just enough to ensure that they are. We define this ideal ADOPS reward as $F' = F + F_2$, where $F$ is some arbitrary initial shaping reward, and $F_2$ is defined as

$$F_2 = \begin{cases} \min(0, V_E^* - Q_E^* + V_I^* - \gamma V_I^*(s') - F - \epsilon) & \text{if} \quad Q_E^* < V_E^* \\ \max(0, V_E^* - Q_E^* + V_I^* - \gamma V_I^*(s') - F) & \text{if} \quad Q_E^* \geq V_E^*. \end{cases} \tag{15}$$

Here, $\epsilon$ is an arbitrarily small positive constant. We are defining $V_I^*$ here such that it represents the *maximum* intrinsic reward achievable while following an extrinsically optimal policy.

The first case of this equation can be intuitively thought of as checking to see if Equation 14 is violated, and if so adjusting the intrinsic reward downwards until it no longer is ("if this action is extrinsically suboptimal, make sure that it is still suboptimal when the IM is taken into account"). Conversely, the second case checks to see if Equation 13 is violated, and adjusts the IM upwards if so until it is not ("if this action is extrinsically optimal, make sure that its intrinsic returns are equal to those of every other extrinsically optimal action").

A full proof that this version of ADOPS preserves the optimal policy can be found in Appendix B.2. In the next section, we introduce a practically-implementable version of ADOPS, and prove that it preserves optimality, as well.

## 5.2. A Practical, Easy-To-Use Form of ADOPS

While it would be ideal, it is usually not feasible to implement Equation (15) , as it requires an accurate estimate of the optimal value function. Let's assume instead that we have access to some critic function that allows us to make approximations of the value function of a given state, and the Q-function of taking some action in that given state, under the agent's current policy $\pi$. Let us also assume that this critic handles the extrinsic and intrinsic rewards separately, such that we can deal with them independently (this is already a common practice, for example in (Burda et al., 2019), whose example we follow in Section 6 ). We notate these estimates as $\hat{V}_E^\pi, \hat{V}_I^\pi, \hat{Q}_E^\pi$, and $\hat{Q}_I^\pi$. This notation has implicit variable-dependence, but occasionally throughout this section we make their variable dependence explicit, as in $\hat{Q}_E^\pi(s, a)$.

Given some initial shaping reward $F$, about which we make no assumptions, we define the ADOPS reward to be

$$F_2 = \begin{cases} \min(0, V_E^\pi - Q_E^\pi + V_I^\pi - \gamma V_I^\pi(s') - F - \epsilon) & \text{if} \quad Q_E^\pi < V_E^\pi \\ \max(0, V_E^\pi - Q_E^\pi + V_I^\pi - \gamma V_I^\pi(s') - F) & \text{if} \quad Q_E^\pi \geq V_E^\pi. \end{cases} \tag{16}$$

We define this shaping reward in terms of the actual value and Q-function values (as opposed to their critic-based estimates) for the proof, and discuss our conditions for implementing this practically at the end of this section.

For our proof, we make an additional assumption about our optimization process itself. Let $\pi_n$ be a policy that takes action $a_n$ in some state $s \in S$. Additionally, take $\pi_m$ to be a policy that is identical to $\pi_n$, except that it takes action $a_m$ in that same state $s$. We will say that a policy $\pi_n$ is "unstable" if there exists a $\pi_m$ such that $Q_{IE}^{\pi_m}(s, a_n) < V_{IE}^{\pi_m}(s)$ and $Q_{IE}^{\pi_n}(s, a_m) \geq V_{IE}^{\pi_n}(s)$. In other words, if a policy is unstable, that means there exists some other extremely similar policy, which differs by only one action in one state, and yet is strictly preferred under any nontrivial mixture of the two policies. Conversely, a policy is "stable" if no such strictly-preferable policy exists. We can now state our assumption:

**Assumption 5.1.** The training algorithm being used will, upon convergence, only execute stable policies.

This assumption can be thought of as a requirement that the learning algorithm being employed will only converge to a policy that is preferred to all other policies in the local policy space (a local optimum). Note that any optimal (with respect to intrinsic and extrinsic rewards) policy will always be stable, and thus that any algorithm guaranteed to converge to an optimal policy is also guaranteed to converge to a stable policy. Intuitively, then, we should expect any competent learning algorithm not to ever converge to an unstable policy, as any exploration around the local policy space would cause the agent to learn action $a_m$ over $a_n$. Indeed, any learning algorithm that does not ever explore the local policy space sufficiently so as to find such a strictly-preferred, small perturbation to its current policy is unlikely to have learned much in the first place. As such, this is a reasonable assumption to make, at least in the limit as training converges to a policy, which is the domain in which we're interested for the purposes of leaving the set of optimal policies unchanged.

We now prove that adding an ADOPS reward term to any initial $F$ preserves optimality.

**Theorem 5.2.** *An intrinsic reward of the form $F' = F + F_2$ with $F_2$ defined according to Equation 16 will preserve the set of optimal policies.*

*Proof.* We proceed by first showing that for all possible actions the agent could take, $Q_{IE}^\pi \geq V_{IE}^\pi$ iff $Q_E^\pi \geq V_E^\pi$, and $Q_{IE}^\pi < V_{IE}^\pi$ iff $Q_E^\pi < V_E^\pi$. We then, from this, show that for all stable policies under Assumption 5.1, $\mathrm{argmax}_a(Q_{IE}^\pi) = \mathrm{argmax}_a(Q_E^\pi)$.

We begin by rewriting Equation (16) to be more concise, and removing the cases, while keeping it mathematically equivalent. We define

$$\Omega = V_E^\pi - Q_E^\pi + V_I^\pi - \gamma V_I^\pi(s') - F \quad (17)$$
$$C_1 = \mathbb{1}(Q_E^\pi < V_E^\pi \wedge \Omega > 0) \quad (18)$$
$$C_2 = \mathbb{1}(Q_E^\pi \geq V_E^\pi \wedge \Omega < 0) \quad (19)$$
$$C_3 = \mathbb{1}(Q_E^\pi < V_E^\pi \wedge \Omega \leq 0) \quad (20)$$

where $C_1, C_2$, and $C_3$ are indicator functions that are 1 if the condition in them is true, and 0 otherwise. We can then rewrite $F_2$ as

$$F_2 = \Omega - (C_1 + C_2)\Omega - C_3\epsilon. \quad (21)$$

We can now rewrite $Q_{IE}^\pi$ as

$$Q_{IE}^\pi \quad (22)$$
$$= \mathop{\mathbb{E}}_{s' \sim P(s'|a,s)} [R + F + F_2 + \gamma V_{IE}^\pi(s')] \quad (23)$$
$$= \mathop{\mathbb{E}}_{s' \sim P(s'|a,s)} [R + V_{IE}^\pi - Q_E^\pi + \gamma V_E^\pi(s') - (C_1 + C_2)\Omega - C_3\epsilon] \quad (24)$$
$$= \mathop{\mathbb{E}}_{s' \sim P(s'|a,s)} [V_{IE}^\pi - (C_1 + C_2)\Omega - C_3\epsilon]. \quad (25)$$

When taking the $\mathrm{argmax}$ of this result, the $V_{IE}^\pi$ term drops out due to having no $a$-dependence, and we are left with

$$\mathrm{argmax}_a Q_{IE}^\pi = \mathrm{argmax}_a \mathop{\mathbb{E}}_{s' \sim P(s'|a,s)} [-(C_1 + C_2)\Omega - C_3\epsilon]. \quad (26)$$

For any given $a$ value, either $Q_E^\pi(a) < V_E^\pi$, or $Q_E^\pi(a) \geq V_E^\pi$. If $Q_E^\pi(a) < V_E^\pi$, then $a$ will not be included in $\mathrm{argmax}_a(Q_E^\pi(a))$. Correspondingly, if $Q_E^\pi(a) < V_E^\pi$, then Equations 17, 18, 19 and 20 require that $C_2 = 0$, and one of either $-C_1\Omega$ or $-C_3\epsilon$ will be nonzero and strictly negative for these actions, with the other term being zero. Therefore, for actions that are extrinsically worse in expectation than the current policy, the term inside the $\mathrm{argmax}$ of Equation 26 will be strictly less than zero. Thus, following Equation 25, if $Q_E^\pi(a) < V_E^\pi$, then $Q_{IE}^\pi(a) < V_{IE}^\pi$.

If $Q_E^\pi(a) \geq V_E^\pi$, however, both the $C_1$ and $C_3$ terms in the $\mathrm{argmax}$ of Equation 26 will be zero. The $C_2$ term will be strictly positive if $\Omega < 0$, and zero otherwise. This observation combined with Equation 25 implies that $Q_{IE}^\pi \geq V_{IE}^\pi$. Therefore all actions with $Q_E^\pi(a) < V_E^\pi$ will not be included in $\mathrm{argmax}_a(Q_{IE}^\pi(a))$, as there will always be at least one action with a greater expected $Q_{IE}^\pi(a)$ (the best action sampled from the current policy $\pi$).

We've now shown that, in any given state $s$, any action in $a^* \in \mathrm{argmax}_a(Q_E^\pi(s, a))$ will have $Q_{IE}^\pi(s, a^*) > V_{IE}^\pi(s)$, but we've not yet proven that each $a^*$ will be in $\mathrm{argmax}_a(Q_{IE}^\pi(s))$. To prove this, let's now assume for contradiction that there exists some other action $\bar{a} \notin \mathrm{argmax}_a(Q_E^\pi(s, a))$ such that $Q_{IE}^\pi(s, \bar{a}) > Q_{IE}^\pi(s, a^*)$. We define $\pi_{\bar{a}}$ to be identical to policy $\pi$ except that it takes action $\bar{a}$ in state $s$. We can then prove that $\pi_{\bar{a}}$ is unstable,

and thus that, under Assumption 5.1, a converged agent will not follow it. Under this new policy, we can calculate the new Q-function of $a^*$:

$$Q_{IE}^{\pi_{\bar{a}}}(s, a^*) = \mathbb{E}_{s' \sim P(s'|a,s)} V_{IE}^{\pi_{\bar{a}}} - (C_1 + C_2)\Omega - C_3\epsilon). \tag{27}$$

Because, by definition, $V_E^{\pi_{\bar{a}}}(s) < Q_E^{\pi_{\bar{a}}}(s, a^*)$ (as otherwise $\bar{a}$ would be an extrinsically optimal action in $s$), we know that both the $C_1$ and $C_3$ terms must equal zero, leaving only the $C_2$ term, which is always positive if nonzero. This proves that $V_{IE}^{\pi_{\bar{a}}} \leq Q_{IE}^{\pi_{\bar{a}}}(s, a^*)$. That is one of the two conditions required for $\pi_{\bar{a}}$ to be unstable.

On the other hand, if the agent follows policy $\pi_{a^*}$, defined as a policy that's identical to $\pi$ except that it takes action $a^*$ in state $s$, the Q-function of taking action $\bar{a}$ becomes

$$Q_{IE}^{\pi_{a^*}}(s, \bar{a}) = \mathbb{E}_{s' \sim P(s'|a,s)} V_{IE}^{\pi_{a^*}} - (C_1 + C_2)\Omega - C_3\epsilon). \tag{28}$$

In this formulation, since we know that $V_E^{\pi_{a^*}}(s) > Q_E^{\pi_{a^*}}(s, \bar{a})$, we know that the $C_2$ term must be zero, while one of either the $C_1$ or $C_3$ terms must be strictly negative, with the other being zero. This implies that $Q_{IE}^{\pi_{a^*}}(s, \bar{a}) < V_{IE}^{\pi_{a^*}}(s)$: the other condition required for $\pi_{\bar{a}}$ to be unstable.

Because action $a^*$ is strictly better than $\bar{a}$ when the agent is following policy $\pi_{a^*}$, and no worse when the agent is following policy $\pi_{\bar{a}}$, it follows that it is strictly better when following any nontrivial mixture of the two policies, and that $\pi_{\bar{a}}$ is an unstable policy. From Assumption 5.1, we know that the learning algorithm will never converge to such a policy. Thus we have a contradiction: in convergence, there exists no policy for which $Q_{IE}^{\pi}(s, \bar{a}) > Q_{IE}^{\pi}(s, a^*)$ that is not in $\text{argmax}_a(Q_E^{\pi})$, and thus every action in $\text{argmax}_a(Q_{IE}^{\pi})$ is also in $\text{argmax}_a(Q_E^{\pi})$.

A similar line of reasoning goes the other direction. Given an action $a^* \in \text{argmax}_a(Q_{IE}^{\pi}(s, a))$ and another action $\bar{a} \notin \text{argmax}_a(Q_{IE}^{\pi}(s, a))$ with $Q_E^{\pi}(s, \bar{a}) > Q_E^{\pi}(s, a^*)$, we find that $Q_{IE}^{\pi_{a^*}}(s, \bar{a}) < V_{IE}^{\pi_{a^*}}$ and $Q_{IE}^{\pi_{\bar{a}}}(s, a^*) \geq V_{IE}^{\pi_{\bar{a}}}$, thus concluding that this $\bar{a}$ is unstable as well.

When dealing with convergent policies then, given Assumption 5.1, we must conclude

$$\text{argmax}_a(Q_{IE}^{\pi}) = \text{argmax}_a(Q_E^{\pi}) \quad \forall \pi. \tag{29}$$

Because we can ensure that the learned value functions and action-value functions using the full reward will induce the same optimal actions comparing to the extrinsic reward for any converged policy, the training process will align with the Bellman optimality equation (Peng & Williams, 1993) using

the extrinsic reward only. Therefore, upon convergence, the underlying learning algorithm using the ADOPS shaped reward will subsequently produce an optimal policy under the extrinsic reward, and Equation 11 holds.

$\square$

This proof so far has been for an $F_2$ wherein we have some access to perfect estimations of state and action values. To extend our framework to function approximations of state and action values, we apply an additional assumption that the base learning algorithm for updating different value estimations convergences. In other words, under the premise that the approximation error is bounded by $\varepsilon$, the value estimations are bounded by a factor $C_\varepsilon$ with regards to the approximation error (Agarwal et al., 2019; Jin et al., 2023). With our proven optimality-preserving property then, the underlying learning algorithm with the shaping reward from ADOPS, in the worst-case scenario, shares the same convergence property as the original version of the problem, without utilizing the ADOPS reward.

## 6. Empirical Results

We test ADOPS, as well as prior optimality-preserving reward shaping methods in the Montezuma's Revenge (Bellemare et al., 2013) Atari Learning Environment (ALE) with RND IM (Burda et al., 2019). We find that PBIM, GRM, and PIES all fail to converge to a policy that outperforms the policy trained on the baseline IM. We tested several versions of ADOPS in this environment and found that all versions tested achieve higher performance than the baseline IM. We provide details of our experiments in Appendix A.1.

### 6.1. PBIM

We find that PBIM with RND, whether normalized or non-normalized, fails to ever obtain nonzero extrinsic rewards in Montezuma's Revenge. PBIM almost immediately saturates the agent's action probabilities in this environment (the average probability of the action being taken). We find that the culprit is PBIM's reliance on accounting for *all* its intrinsic rewards at the end of the episode, combined with Montezuma's Revenge's long episode lengths. This is due to the exponential nature of the denominator in the final reward for an episode under PBIM:

$$F'_{N-1} = -\frac{U_0}{\gamma^{N-1}}, \tag{30}$$

where $U_0$ is the discounted return of the previous intrinsic rewards obtained throughout the episode. When training in smaller environments, such as the cliff walker and grid world experiments of (Forbes et al., 2024a), this reward is reasonable, and effectively speeds up training. In environments like Montezuma's Revenge, however, with potentially very

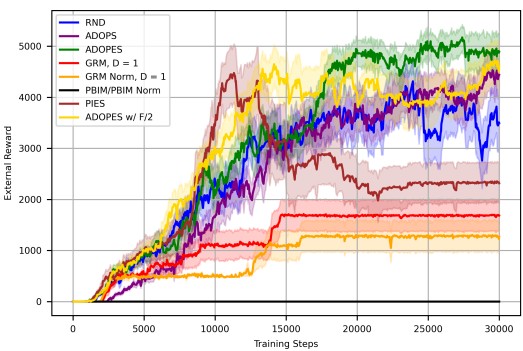

*Figure 1.* Comparison of all methods in Montezuma's Revenge. All plots are smoothed over the past 10 steps. Error bars are errors on the mean. Differences between RND and both normalized and non-normalized GRM are statistically significant, with $p = 0.009$ and $p = 0.031$, respectively. Differences between RND and PIES are nearly, but not quite statistically significant, with $p = 0.059$. Differences between normalized and non-normalized GRM are not statistically significant, with $p = 0.37$ (two-sided t-test). ADOPES is better than RND to a statistically significant degree, $p = 0.038$. ADOPS, ADOPES, and ADOPES w/ $F/2$ all improve on PIES to a statistically significant degree, with $p = 4.4e - 5$, $p = 2.4e - 6$, and $p = 6.4e - 5$, respectively (two-sided t-test). They similarly improve over GRM and PBIM. $N = 10$ for each GRM run, $N = 1$ for each PBIM run, and $N = 20$ for all other methods.

long episodes and relatively low $\gamma$ values, the denominator in this reward becomes unwieldy, and the end-of-episode 'penalty' term exponentially explodes, preventing further learning. We investigate and discuss this finding further in Appendix A.2.

### 6.2. GRM

To compare our method to GRM, we chose the same subset of GRM methods recommended in (Forbes et al., 2024b), parameterized by the delay hyperparameter $D$ in Equation (8). To determine for which value for $D$ GRM fares best in this environment, we tested single runs across a wide range of $D$ values. We detail the results of this hyperparameter test in Appendix A.3. We find that lower $D$ values are consistently better, keeping in line with our results from Section 6.1, and further suggesting that too-long delays between when an intrinsic reward is received by the agent and accounted for can cause an exponential explosion. Our best-performing candidate used $D = 1$. We therefore tested both normalized and non-normalized versions of this equation as our representative tests of GRM.

As can be seen from Figure 1, all versions of GRM or PBIM failed to reach the same average cumulative extrinsic reward as RND, to a statistically significant degree. Additionally, non-normalized GRM seems to outperform normalized

GRM. Though this difference isn't statistically significant, we hypothesize that it is because our environment's extrinsic reward is strictly positive, which tends to make longer episodes more likely to have extrinsically higher returns, and thus a bias toward prolonging the episode like that described in (Forbes et al., 2024a) is not harmful in the same way as in their experiments, which are in environments where the optimal policy is to reach a goal state as quickly as possible. Thus, the normalization's effect of mitigating this bias may not be as useful here.

### 6.3. PIES

We tested PIES with a $\zeta$ decay rate $\frac{1}{C} = \frac{1}{15000}$. We chose this rate to strike a balance between giving the agent enough time with intrinsic rewards to explore the environment and giving it enough time without intrinsic rewards to converge to an optimal policy. With this decay rate, PIES "starts" conserving the optimal policy, in theory, at exactly the halfway point in training, as this is the point at which it begins to return an IM of zero. We also modified the PIES algorithm slightly to suit our larger training environment: we update the value of the $\zeta$ coefficient every iteration, rather than every episode, as we are training multiple agents in parallel, and with highly variant episode lengths from one episode to another. We also modified PIES by using it to shape the coefficient of IM, rather than a potential-based Markovian shaping reward, as it was used in (Behboudian et al., 2022): we consider this itself to be a novel extension of the PIES methodology, albeit a simple one.

Our PIES results are also plotted in Figure 1. Note that, while PIES performs well initially, it decreases in performance rapidly upon approaching the halfway point of training, and never recovers: in other words, its performance worsens as soon as PIES begins to approach conserving the optimal policy set of the underlying environment. This suggests a clear trade-off: PIES can either conserve the optimal policy of the underlying environment, or allow for IM to be used effectively, but struggles to do both simultaneously. Also of note is that the multiplicative coefficient for the IM we used for our RND runs (and thus the effective value of $\zeta$ at the first iteration for our PIES runs) was simply 1, as in the original RND paper (Burda et al., 2019). As that paper didn't test different values for this hyperparameter, the initially good performance of PIES in Figure 1 suggests that the best value of this coefficient is likely lower than the default value used. In the first half of Figure 1, after all, PIES is essentially equivalent to RND as implemented in the same plot, but with a lower and steadily decreasing multiplicative coefficient for the intrinsic rewards. In Section 6.4, we present more results that suggest this is indeed the best explanation for PIES's good early performance.

## 6.4. ADOPS

We test three versions of our method. The first, our baseline ADOPS method, simply implements Equation (16) using the critic networks' estimations of the relevant quantities.

Noting that these critics' estimations are much better later on in training than earlier, we also implement a fusion of our method with PIES, which we call Action Dependent Optimality Preserving Explicit Shaping (ADOPES). It is equivalent to ADOPS, except with a $\zeta$ coefficient multiplied by $F_2$ that begins at $0$, then scales *up* throughout training, to a final value of $1$, rather than down. This means that PIES and ADOPES both begin identically, giving the agent unfettered access to the base IM at the start of training. However, while PIES linearly scales this reward down throughout training until the agent is receiving no IM at all, ADOPES instead scales up the coefficient by which $F_2$ is multiplied, thus preserving the optimal policy in a way that still gives the agent access to IM during the latter half of training.

Finally, inspired by the early success of PIES in Figure 1, to see what extent lowering the starting multiplicative coefficient for IM had on the training speed, we also trained on a version of ADOPES with the starting IM coefficient lowered to .5, or half of what it was in the original RND paper or our other experiments.

All our ADOPS-based runs are included in Figure 1. All of our methods outperform all prior optimality-preserving work in terms of the average extrinsic return of the final policy to a statistically significant degree, and ADOPES also outperforms RND to a statistically significant degree, as well as being the best-performing method overall. Our modified version of ADOPES with a lowered IM coefficient learned more quickly than the other runs, supporting our explanation that PIES's promising results earlier on in training were the result of incidentally finding better values for the IM scaling coefficient along the way to zero. However, unlike PIES, ADOPES allows the agent to keep receiving intrinsic rewards past the halfway point of training and thus can maintain its ability to receive higher extrinsic returns throughout training.

## 7. Conclusion

We present ADOPS, a novel plug-and-play method for implementing shaping rewards while preserving the optimal policy of the underlying environment. We prove this optimality-preserving property, as well as the lack of several previously-necessary assumptions, and demonstrate SOTA performance in a difficult environment, as compared with other optimality-preserving methods.

## Acknowledgments

Thanks to Nick Gauthier for help setting up and debugging the relevant python packages.

## Impact Statement

This paper presents work whose goal is to advance the field of Machine Learning. There are many potential societal consequences of our work, none which we feel must be specifically highlighted here.

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

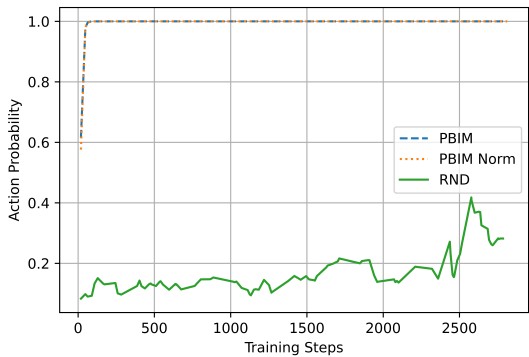

*Figure 2.* Action probabilities for RND and PBIM.

## A. Experimental Details

Here, we detail more about our experiments.

### A.1. Experimental Setup

Montezuma's Revenge ALE environment is a challenging sparse-rewards environment with a history of being used as a benchmark for IM methods (Burda et al., 2018; 2019). We used the environment unmodified. Our base model for all trained agents was the PPO algorithm (Schulman et al., 2017), with additional intrinsic rewards from the RND network (Burda et al., 2019). We use the same hyperparameters as the 32-worker Convolutional Neural Net (CNN) runs in (Burda et al., 2019), with one key difference: while (Burda et al., 2019) clips external rewards to the interval $[-1, 1]$, we train on the external reward as-is because we are primarily concerned with methods that preserve the optimal policies of the underlying environment, and this clipping might alter that set (for example, by eliminating the effective difference between a reward of $1,000$ and $2,000$, and thus changing whether it is strictly preferred for the agent to collect the latter reward first). To compensate for this change, we scaled the extrinsic reward down by a factor of $1000$, to keep the expected order of magnitude of extrinsic rewards identical to that in (Burda et al., 2019).[4] For methods requiring normalization, such as PBIM Norm, we used exponential smoothing with $\alpha = 0.05$. For all ADOPS variants, we used $\epsilon = 1e - 7$.

We use the implementation in PPO-RND (Kazemipour, 2022) as an initial codebase. PPO-RND does not have an active license. We conducted experiments using gymnasium API simulated by Arcade Learning Environment (ALE) (Bellemare et al., 2013) for Montezuma's Revenge. We use "ALE/MontezumaRevenge-v4" in our experiments. Gymnasium API is under the MIT license and ALE is under GPL-2.0 license.

We conducted our experiments on two servers with Ubuntu 22.04. One server has 12 Intel(R) Xeon(R) CPU E5-1650, 2 NVIDIA GeForce GTX 1080, and 32 GiB memory. We run two experiments concurrently on it. The other server has 12 AMD EPYC 7401P CPU, 1 NVIDIA TITAN RTX, and 30 GiB memory. Each run takes around 30 hours.

### A.2. Failure of PBIM To Obtain Rewards

As can be seen in Figure 2, PBIM and PBIM Norm immediately saturate the action probability, preventing the agent from ever obtaining nonzero extrinsic rewards. This is contrasted with action probabilities for a run of RND, which demonstrates a typical action probability for successful methods in this environment at the beginning of training. As the PBIM methods immediately converge to an action probability of 1, they never explore the environment, and thus never obtain any extrinsic rewards.[5] We use the same maximum episode length and intrinsic discount $\gamma_I$ values as (Burda et al., 2019), which are $4,500$ and $.99$, respectively. As an order-of-magnitude calculation, $\frac{1}{.99^{4500}} \approx 10^{19}$, implying that we should expect the intrinsic rewards in the final time steps of the longest episodes in this environment with PBIM to be around this large. Indeed,

---

[4]We found that without this scaling factor, the agent performed significantly worse. This scaling factor is considered separately from the scaling factor discussed in Section 6 in the context of PIES and ADOPES, as it is for extrinsic rewards, whereas that factor is for IM.

[5]In both cases, the particular action being taken at every time step by the agent is to not move at all.

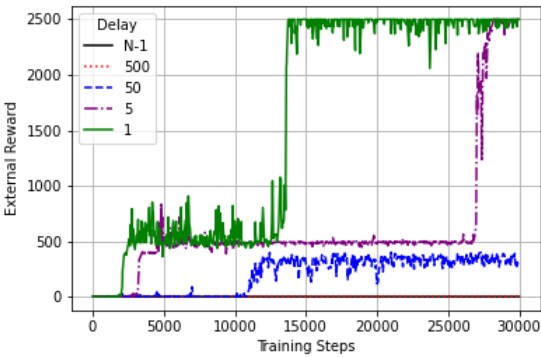

*Figure 3.* Extrinsic returns for runs with various values of $D$ in Equation (8). All runs are normalized. The run with $D = N - 1$ is equivalent to PBIM Norm from (Forbes et al., 2024a).

through inspection, we did observe rewards of this magnitude under both PBIM and PBIM-Norm, and it was these rewards that saturated the action probability in Figure 2 and thus prevented learning. Thus, we find that neither PBIM method is suitable to be used in environments with potentially long episode lengths and low $\gamma$ values, but should instead be limited to environments with shorter episodes and long time horizons within those episodes.[6]

### A.3. Hyperparameter Test For D in GRM

We plot the results of our GRM $D$ hyperparameter test in Figure 3. We ran these tests using the normalized versions of GRM, taking into account the results of (Forbes et al., 2024b) suggesting that this makes a wider range of $D$ values robust against introducing unwanted biases from the data.

## B. Additional Proofs

Here, we detail additional proofs.

### B.1. Proof Regarding GRM's Lack of Generality

Here, we prove that there is a relevant set of optimality-preserving shaping functions that cannot be accommodated by GRM or PBRS.[7]

**Theorem B.1.** *There exist optimality-preserving reward shaping functions which cannot be written as GRM shaping functions.*

*Proof.* In a given MDP, take the shaping function $F'_t = R_t$. Trivially, this preserves the optimal reward. For a proof by contradiction, we assume there exists some $m_{t,t'}, F_t$ such that this can be written as a GRM shaping reward in the form of Equation (7). Taking the intrinsic return at time $t$, we get

$$U_t^I = \sum_{j=t}^{N-1} \gamma^{j-t} F'_j \tag{31}$$

$$= \sum_{j=t}^{N-1} \gamma^{j-t} F_j - \sum_{j=t}^{N-1} \sum_{i=0}^{j} \gamma^{i-t} F_i m_{j,i}, \tag{32}$$

where $U_t^{I_{old}}$ is the cumulative discounted sum of $F_t$. We know from the proof for GRM's optimality in (Forbes et al., 2024b)

---

[6] For reference, the longest episode length in an environment wherein PBIM successfully sped up training performance in (Forbes et al., 2024a) was 640 time steps, and the discount factors tested ranged from .99 to .995.

[7] These are equivalent sets of functions, as proved in (Forbes et al., 2024b).

that this entails

$$U_t^I = -\sum_{i=0}^{t-1} \gamma^{i-t} F_i + \sum_{i=0}^{t-1} \gamma^{i-t} F_i \left(\sum_{j=i}^{t-1} m_{j,i}\right). \tag{33}$$

Since all $F_{t'}$ must be future-agnostic (Equation (5)), we can apply Equation (5) to assert that $U_t^I$ has no $a_t$-dependence. But this is not generally true, and certainly false in any nontrivial environment: because we've defined $F_t'$ such that $U_t^I = U_t^E$, this would imply that in every environment, the agent actions cannot influence its return. We have thus derived a contradiction, and so our assumption that GRM can accommodate $F_t' = R_t$ was false. $\square$

To further understand the implications of this proof, examine what would happen if we were to plug this counterexample shaping function $F_t' = R_t$ into the GRM framework: it would convert it into a shaping reward that is, indeed, action-independent. In doing so, it would discard much of the useful information contained in that reward.

In contrast with GRM or other PBRS-based methods, ADOPS can easily accommodate reward shaping functions of this type: in fact, the theoretical best ADOPS method we define in Equation (15) correctly identifies this shaping function as not disrupting the optimal policy to begin with, and returns it back unchanged, with $F_t' = F_t$.

## B.2. A Proof of Optimality of the "Ideal" ADOPS Shaping Function

Given some initial shaping reward $F$, about which we make no assumptions, we define the ideal (usually inaccessible) ADOPS shaping reward to be

$$F_2 = \begin{cases} \min(0, V_E^* - Q_E^* + V_I^* - \gamma V_I^*(s') - F - \epsilon) & \text{if} \quad Q_E^* < V_E^* \\ \max(0, V_E^* - Q_E^* + V_I^* - \gamma V_I^*(s') - F) & \text{if} \quad Q_E^* \geq V_E^*. \end{cases} \tag{34}$$

Here, we are defining $V_I^*$ such that it represents the *maximum* intrinsic reward achievable while following an externally optimal policy. While for any extrinsically optimal policy, it will always be true that $V_E^*$ is equivalent to that of any other extrinsically optimal policy, they may differ in terms of which achieves higher intrinsic reward: when this is the case, we take $V_I$ to be the maximum of these.

We prove that adding this reward term to any initial $F$ preserves optimality.

**Theorem B.2.** *An intrinsic reward of the form $F' = F + F_2$ with $F_2$ defined according to Equation 34 will preserve the set of optimal policies.*

*Proof.* Proof by contradiction. To prove that optimality is preserved, it suffices to prove that Equations 13 and 14 hold for any trajectory. Let us assume that there exists some $\bar{\pi}$ such that

$$Q_E^{\bar{\pi}} < V_E^* \qquad Q_E^{\bar{\pi}} + Q_I^{\bar{\pi}} \geq V_E^* + V_I^*. \tag{35}$$

This is equivalent to assuming that Equation 14 does not hold. We can then write the quantity $Q_E^{\bar{\pi}} + Q_I^{\bar{\pi}}$ as

$$Q_E^{\bar{\pi}} + Q_I^{\bar{\pi}} = Q_E^{\bar{\pi}} + \mathbb{E}(F + F_2 + \gamma V_I^{\bar{\pi}}(s')). \tag{36}$$

From the first component of our assumption, we know that $F_2$ is going to be defined according to the topmost case of Equation 34. We can then decompose our expression into contributions from terms wherein $F_2 = 0$ (from the min function), and those in which it does not. We can define

$$Q_E^{\bar{\pi}} + Q_I^{\bar{\pi}} \tag{37}$$
$$= P_0(Q_E^{\bar{\pi}} + Q_I^{\bar{\pi}}) + (1 - P_0)(Q_E^{\bar{\pi}} + Q_I^{\bar{\pi}}) \tag{38}$$
$$= P_0(Q_E^{\bar{\pi}} + \mathbb{E}(F_0 + \gamma V_{I,0}^{\bar{\pi}}(s'))) \tag{39}$$
$$+ (1 - P_0)(Q_E^{\bar{\pi}} + \mathbb{E}(F_{\neq 0} + F_{2,\neq 0} + \gamma V_{I,\neq 0}^{\bar{\pi}}(s'))) \tag{40}$$

where $P_0$ is a scalar representing the proportion of trajectories from the relevant state in the relevant action in which $F_2 = 0$, and the subscripts $0$ and $\neq 0$ represent the contributing terms in which $F_2 = 0$ and $F_2 \neq 0$, respectively.

For the terms with the $\neq 0$ subscript, we find

$$Q_E^{\bar{\pi}} + \mathbb{E}(F_{\neq 0} + F_{2,\neq 0} + \gamma V_{I,\neq 0}^{\bar{\pi}}(s')) \tag{41}$$

$$=Q_E^{\bar{\pi}} + \mathbb{E}(F_{\neq 0} + V_E^* - Q_E^{\bar{\pi}} + V_I^* - \gamma V_{I,\neq 0}^{\bar{\pi}}(s') - F_{\neq 0} - \epsilon + \gamma V_{I,\neq 0}^{\bar{\pi}}(s')) \tag{42}$$

$$=Q_E^{\bar{\pi}} + \mathbb{E}(V_E^* - Q_E^{\bar{\pi}} + V_I^* - \epsilon) \tag{43}$$

$$=V_E^* + V_I^* - \epsilon. \tag{44}$$

We also know from the condition in 34 that

$$V_E^* - Q_E^{\bar{\pi}} + V_I^* - \gamma V_{I,0}^{\bar{\pi}}(s') - F_0 - \epsilon \geq 0 \tag{45}$$

$$Q_E^{\bar{\pi}} + \gamma V_{I,0}^{\bar{\pi}}(s') + F_0 \leq V_E^* + V_I^* - \epsilon \tag{46}$$

$$Q_E^{\bar{\pi}} + \mathbb{E}(\gamma V_{I,0}^{\bar{\pi}}(s') + F_0) \leq V_E^* + V_I^* - \epsilon, \tag{47}$$

which gives us an upper bound for the terms with the 0 subscript.

Taking our assumption, then, we can replace both components of Equation 40 with their respective reductions, which gives us

$$V_E^* + V_I^* \leq Q_E^{\bar{\pi}} + Q_I^{\bar{\pi}} \tag{48}$$

$$\leq P_0(V_E^* + V_I^* - \epsilon) + (1 - P_0)(V_E^* + V_I^* - \epsilon) \tag{49}$$

$$\leq V_E^* + V_I^* - \epsilon. \tag{50}$$

This is a contradiction, and thus we have proven that Equation 14 holds.

To prove Equation 13, we similarly assume that there exists some extrinsically optimal $\pi^*$ such that

$$Q_E^{\pi^*} = V_E^* \qquad Q_E^{\pi^*} + Q_I^{\pi^*} \neq V_E^* + V_I^*. \tag{51}$$

The second component of our assumption includes two possible cases:

$$Q_E^{\pi^*} + Q_I^{\pi^*} < V_E^* + V_I^* \quad \text{or} \quad Q_E^{\pi^*} + Q_I^{\pi^*} > V_E^* + V_I^*. \tag{52}$$

Let's begin by assuming the first of these. We can again decompose $Q_E^{\pi^*} + Q_I^{\pi^*}$ into two parts, using the constant $P_0$ to denote the percentage of contributing trajectories that are set to zero:

$$Q_E^{\pi^*} + Q_I^{\pi^*} \tag{53}$$

$$=P_0(Q_E^{\pi^*} + Q_I^{\pi^*}) + (1 - P_0)(Q_E^{\pi^*} + Q_I^{\pi^*}) \tag{54}$$

$$=P_0(Q_E^{\pi^*} + \mathbb{E}(F_0 + \gamma V_{I,0}^{\pi^*}(s'))) \tag{55}$$

$$+(1 - P_0)(Q_E^{\pi^*} + \mathbb{E}(F_{\neq 0} + F_{2,\neq 0} + \gamma V_{I,\neq 0}^{\pi^*}(s'))). \tag{56}$$

Here, due to our differing assumption, whether or not a contributing trajectory has its $F_2$ term set to zero is determined by the max function in the latter case of Equation 34, rather than the min function in the former. For trajectories where $F_2 \neq 0$, then, we get

$$Q_E^{\pi^*} + \mathbb{E}(F_{\neq 0} + F_{2,\neq 0} + \gamma V_{I,\neq 0}^{\pi^*}(s')) \tag{57}$$

$$=Q_E^{\pi^*} + \mathbb{E}(F_{\neq 0} + V_E^* - Q_E^{\pi^*} + V_I^* - \gamma V_{I,\neq 0}^{\pi^*}(s') - F_{\neq 0} + \gamma V_{I,\neq 0}^{\pi^*}(s')) \tag{58}$$

$$=Q_E^{\pi^*} + \mathbb{E}(V_E^* - Q_E^{\pi^*} + V_I^*) \tag{59}$$

$$= V_E^* + V_I^*. \tag{60}$$

For conditions with the 0 subscript, we know from the condition in Equation 34 that

$$V_E^* - Q_E^{\pi^*} + V_I^* - \gamma V_{I,0}^{\pi^*}(s') - F_0 \leq 0 \tag{61}$$

$$Q_E^{\pi^*} + \gamma V_{I,0}^{\pi^*}(s') + F_0 \geq V_E^* + V_I^* \tag{62}$$

$$Q_E^{\pi^*} + \mathbb{E}(F_0 + \gamma V_{I,0}^{\pi^*}(s')) \geq V_E^* + V_I^*. \tag{63}$$

Looking first at the first decomposition of the $\neq$ in Equation 52, we get

$$V_E^* + V_I^* > Q_E^{\pi^*} + Q_I^{\pi^*} \tag{64}$$
$$> P_0(V_E^* + V_I^*) + (1 - P_0)(V_E^* + V_I^*) \tag{65}$$
$$> V_E^* + V_I^*. \tag{66}$$

This is a contradiction. The other decomposition of our assumption in Equation 52 implies

$$V_E^* + V_I^* < Q_E^{\pi^*} + Q_I^{\pi^*} \tag{67}$$
$$V_E^* + V_I^* < V_E^* + Q_I^{\pi^*} \tag{68}$$
$$V_I^* < Q_I^{\pi^*}. \tag{69}$$
$$\tag{70}$$

Here, between Equations 67 and 68, we applied our first assumption in Equation 51. Because we have defined $V_I^*$ to be the highest possible intrinsic reward among extrinsically optimal trajectories, this is a contradiction as well. Thus Equation (13) must hold. This suffices to prove that the set of optimal policies remains unchanged. $\qquad\square$

