# OpenReview forum: "Action-Dependent Optimality-Preserving Reward Shaping"
_ICML.cc/2025/Conference — ICML 2025 poster_

### Official Review · Reviewer_cdE4 · 2025-03-09

**Overall Recommendation:** 3

**Summary:**

The paper proposes a new reward-shaping framework that is action-dependent while preserving the optimal policies. The proposed method can convert any intrinsic rewards that are not optimality preserving to preserve optimal policies. Through experiments in Montezuma's Revenge, the authors demonstrate the effectiveness of the proposed method.

**Claims And Evidence:**

The proposed method can be supported by its theories but not sufficient enough by its experiments since only one environment is tested.

**Essential References Not Discussed:**

The following baseline in intrinsic motivation works should be included for a fair comparison,
- [Automatic Intrinsic Reward Shaping for Exploration in Deep Reinforcement
Learning](https://proceedings.mlr.press/v202/yuan23c/yuan23c.pdf) in ICML 23 and other baselines included in this paper.

**Experimental Designs Or Analyses:**

The experiment results on Montezuma's Revenge, while being good empirical evidence for the proposed method, might not be sufficient enough to convince the broader community. There are many other hard-to-explore environments. For example, a more basic giant grid world maze could suffice. Extending such a grid world to a continuous maze is also interesting to see.

**Methods And Evaluation Criteria:**

The proposed method makes sense that the classic PBRS is indeed a sufficient not necessary condition for preserving optimality after shaping. However, the evaluation criteria are a bit less convincing that the method is only evaluated in one single environment.

**Other Comments Or Suggestions:**

N/A

**Other Strengths And Weaknesses:**

- The writing could be improved in the sense that (1) the introduction and background on reward shaping are overly extended, and (2) a more intuitive explanation and examples could be used to illustrate the correctness of theorem 5.3

**Questions For Authors:**

- Is it possible to have the same results without assumptions 5.1, 5.2? If possible, could you characterize the error your approach could induce under imperfect value estimates and imperfect state/action visitations?
- In your proposed shaping functions, what is the input to $Q^*_{I,t+1}$? Is it the state action pair in the next time step? If so, how should we have access to it when we are still calculating the reward for $s_t,a_t$?

**Relation To Broader Scientific Literature:**

The paper extends the PBRS framework by adding action-dependent terms.

**Theoretical Claims:**

Under the assumptions, the overall proof seems legit since I haven't checked every step in detail. But those assumptions look way too strong to be realistic.
- For assumption 5.1 which assumes the neural network estimates of the values are the true values. As the authors claim, if trained long enough, it will converge to true values, but this already hinders the initial motivation of using reward shaping - to train agents sample efficiently.
- For assumption 5.2, if we already assume sufficient exploration of all possible actions in every state vistable by Sn, why do we even need intrinsic motivations? In other words, the intrinsic motivation is designed to motivate the agent to explore possible actions/states. Having this assumption renders intrinsic motivations useless.
- Follow up on assumption 5.2, can you define formally the set of Sn? If we assume sufficient exploration of the action space for every policy, is it the case that Sn will be the same for everyone?

---

> ### Author Rebuttal · Authors · 2025-04-01
>
> Thank you for your time and effort in giving detailed feedback; we really appreciate it.
>
> We’re excited to evaluate our method on additional environments in future work, and believe that these initial results stand as a significant contribution, particularly when paired with our theoretical results. See our response to Reviewer 4wwG for a more detailed overview of our reasons for believing that this work stands on its own as a novel and significant contribution.
>
> You make multiple insightful comments about our theory, and we’ve reworked and rephrased Section 5.2 substantially for clarity and precision in light of them. In particular, we’ve improved the wording of both of our assumptions, which as previously phrased were both more restrictive than necessary, and worded somewhat confusingly. A reworked version of Section 5.2 (our main proof) can be found here, for you to peruse.
>
> https://drive.google.com/file/d/1XTZ3VSuIDFaqHM_NxsjBpQEhkR6DFOyM/view?usp=sharing
>
> We will now address your comments in more detail, in the order that they appear.
>
> For Assumption 5.1: we want to disambiguate between the two goals of ADOPS/of optimality-preserving reward shaping in general: firstly, speeding training/increasing sample efficiency, and secondly preventing reward hacking by preserving the optimal policy. The latter of these goals is what we prove in Theorem 5.3, and what we require our assumptions for. The former goal then, of speeding training, can still be met (and, empirically is met, better than RND itself, to a statistically significant degree) before the point at which Assumption 5.1 is true. Indeed, RND itself has no such theoretical guarantees of conserving the optimal policy, but has sped up training across a wide variety of domains, at the risk of making the agent susceptible to reward hacking. Our theorem is designed to prevent this reward hacking, while leaving the underlying training-speeding qualities of the base IM as unchanged as possible ($F^2=0$ very frequently on any given timestep).
> We have removed the assumption 5.1 and adopted the convergence property of the underlying algorithm. Since the ADOPS ensures that the shaped reward will produce a consistent policy with the extrinsic reward, the learning algorithm would share its convergence property of using the extrinsic reward in the worst case scenario.
>
> We have changed our Assumption 5.2 to be much more specific and limited of an assumption. Namely, it defines a notion of “unstable” policies, which are those for which an extremely similar, locally better policy exists, and so which any competent learning algorithm will quickly leave in favor of a better policy. It’s detailed more precisely in the attached document above, and we believe it to be much improved in both rigor and scope to the original assumption.
>
> We’ve added the baseline you reference to our related work section, and are excited to test against this (and other IM baselines) in future work. We’ve also rewritten Theorem 5.3, in a way that clarifies many of the derivations and gives an intuitive overview at the beginning, and signposts throughout.
>
> As for your questions:
>
> We have substantially reworked our assumptions, and given some sense of what were to happen were they violated.
>
> Your question about the variable dependence of $\gamma Q_{It+1}$ is valid, and brings to light the needless complexity of our previous notation. $\gamma Q_{It+1}$ simply represents all components of $Q_{I}$ that come from a timestep later than $t$. This is equivalent to, and can be more simply written as, $\gamma V_{I}(s’)$. We in fact rely on this equivalence in our paper, in the step between Equations (26) and (27). For clarity, we have replaced all instances of $\gamma Q_{It+1}$ with $\gamma V_{I}(s’)$. This depends, as you might imagine, on $s’$.
>
> Thank you again for your in-depth comments and suggestions; we believe that incorporating them has substantially improved our paper.

---

> > ### Comment · Reviewer_cdE4 · 2025-04-03
> >
> > I thank the authors for the clarifications and the efforts in uploading the revised section. Some of my concerns have been addressed and I will explain what is left in the sequel.
> >
> > Theories in its current form looks reasonable to me. However, removing assumption 5.1 only covers instead of solving the intrinsic limitation of the proposed method, which is, how can we have access to a reasonable value/q-value critique before we even train the agent? From the experiment section, the authors use some preexisting critiques. I wonder how the performance of those critiques are? If they are already good value estimations on themselves, why should we train another agent? If they are not good (far from the true value) violating the formulation in theories, how would you explain the good performance?
> >
> > **Update after rebuttal comment reply**: Thanks for the clarifications, I have increased the score to 3.

---

> > > ### Author Response · Authors · 2025-04-04
> > >
> > > We’re glad to hear some of your concerns were addressed. Thank you for the opportunity to clarify further, as we believe there may be an important misunderstanding here, due to some bad wording in the initial draft of our paper.
> > >
> > > We believe that when you mention our use of “preexisting critiques,” you’re referring to Section 6.4, where we say we’re using the “preexisting network critics’ estimations of the relevant quantities.” This was poor wording on our part, and we’ve amended the paper to be more clear. We address this clarification further in our response to Reviewer PXoU, but in short, we just meant that these critics are already used in the base PPO algorithm, and so our method doesn’t require any additional network architecture. To clarify, $\textbf{we do not use any pre-trained networks or transfer learning of any kind.}$ All of the networks used in our method are randomly initialized, and trained from scratch, just as they would be in a standard PPO algorithm. If you meant something else by the term preexisting critiques, please clarify further so we can address your concerns.
> > >
> > > We would like to emphasize that ADOPS as implemented in our experimental section and in Section 5.2 require only that the Q and V estimations for the $\textit{current}$ policy be accurate, rather than the optimal Q and V values (this is the main advantage that Equation (18) has over Equation (15)). Right as training begins, the critic networks immediately begin receiving good, on-distribution data for the Q/V values of taking the current (randomly initialized) policy, and thus should quickly approximate those before the agent begins performing well at all. In fact, in PPO, the policy section of the network never “sees” any of the rewards themselves, but only the critic networks’ estimations of the Q/V functions for the current policy, which are themselves trained on the base reward. So effective training of the policy to maximize the rewards can’t really even begin unless the critic networks are at least somewhat reliable estimations of the V functions under the current policy. This is true in baseline RND, or even in the case where you’re training without any intrinsic reward, so long as the method you’re using uses a critic network.
> > >
> > > In other words, the accurate estimation of V of the current policy which we require is much different, and much less useful, than an accurate estimation of V of the optimal policy: this is why we still need to train, as training improves the policy network’s performance, and then the critics update to estimate V of the new, improved policy. This happens iteratively throughout training, until the V estimations eventually approach V of an optimal policy.
> > >
> > > Please let us know if you have any further questions or clarifications. Thank you again for your time.

---

### Official Review · Reviewer_PXoU · 2025-03-14

**Overall Recommendation:** 3

**Summary:**

This paper presents Action-Dependent Optimality Preserving Shaping (ADOPS), a technique that transforms intrinsic rewards into a format that maintains optimality and enhances the efficacy of intrinsic motivation in the challenging, reward-sparse environment of Montezuma’s Revenge. Additionally, it establishes that ADOPS supports reward shaping functions that are not aligned with potential-based frameworks: whereas PBRS-based methods necessitate that the cumulative discounted intrinsic return remain independent of actions, ADOPS allows these returns to be contingent upon the agent’s actions while still preserving the set of optimal policies.

**Claims And Evidence:**

The evidence provided in this paper vaguely supports the claims made, raising concerns about the robustness and reliability of the findings. More discussion and claims is needed.

**Essential References Not Discussed:**

no

**Experimental Designs Or Analyses:**

Yes

**Methods And Evaluation Criteria:**

Yes. However, only one environment is tested

**Other Comments Or Suggestions:**

Minor:
Equation (3) should be F_t(s) =γΦ_{t+1}(s')−Φ_t(s) or F'_t(s) =γΦ_{t+1}(s')−Φ_t(s)? However you never define the F' before.
π*_1 and π*_2 should be defined in (13)
what is bar{a}? in (14)?

**Other Strengths And Weaknesses:**

Strengths:
The literature review is well-executed, and the discussion of prior work is thorough.

The theoretical foundations appear to be robust, and the experimental results are also commendable.

Weakness:
The assumption are quite strong: The assumption requires that the estimator functions receive an unbounded amount of unbiased data from the environment under the current policy. In practice, data is always finite and can be biased due to exploration strategies or environmental nonstationarities.

Only one test environment is not enough to demonstrate effectiveness

**Questions For Authors:**

In the claim you presented, could you clarify the specific assumptions you've set aside, such as the requirement for the environment to be episodic and for the intrinsic motivation (IM) to be "future-agnostic"? It appears that these points were not revisited later in the paper. The primary focus of the document seems to be on resolving issues arising from the discrepancy between the action-independent value function and the action-dependent Q-function.

Additionally, you mentioned "using the preexisting network critics' estimations of the relevant quantities"—does this critic remain static, or does it evolve over time as it learns within the ADOPS framework?

**Relation To Broader Scientific Literature:**

The key contributions of the paper focus on encouraging agent exploration in sparse-reward environments.

**Theoretical Claims:**

Yes

---

> ### Author Rebuttal · Authors · 2025-04-01
>
> Thank you for your time, energy, and constructive feedback.
>
> We agree with your assessments about the quality of our empirical evaluations. Our primary contribution is the initial proposal of ADOPS, as well as our theoretical proofs of both optimality and increased generality over prior methods. Our future work will focus on further empirical evaluation of ADOPS in more environments as well as against other IM methods. See our response to Reviewer 4wwG for a more detailed overview of our reasons for believing that this work stands on its own as a novel and significant contribution.
>
> Our assumptions as originally stated were somewhat stronger than they had to be in order for our proof to function, and our initial discussion around them lacked clarity. In light of your helpful feedback, as well as that of other reviewers, we’ve revised them substantially. We dropped assumption 5.1 and simply adopted the convergence property of the base learning algorithm. Additionally, our previous Assumption 5.2 (now 5.1) requires essentially that the learning algorithm executes locally greedy actions in the policy space, and we describe when it’s introduced why this is a reasonable assumption. We’ve also substantially reworded sections of the proof of Theorem 5.3, for clarity. We’ve attached an anonymous revised text of Section 5.2, so that you can examine these changes holistically.
>
> https://drive.google.com/file/d/1XTZ3VSuIDFaqHM_NxsjBpQEhkR6DFOyM/view?usp=sharing
>
> To generally address your comment about requiring enough data for the critic estimators to converge to their theoretical values: while we’ve altered our assumption here, this critique still touches on an important point we’d like to address. We’re building on a prior line of work in optimality-preserving reward shaping, including (Ng, 1999), (Devlin 2012), and the PIES, PBIM, and GRM frameworks among others, that all are based on preserving the optimal policy set of the underlying environment. The implicit assumption behind this framework is that the agent being trained on an optimality-preserving reward function will converge to an optimal, or near-optimal policy, and avoid the alternative, which is to hack the shaping reward. After all, if the agent being trained does not converge to an optimal policy, or a reasonable-enough approximation of it, then practically, the importance of a shaping function being optimality-preserving is somewhat limited. So some assumption like “the agent will eventually converge to an optimal or near-optimal policy” is implicit in all prior work in this area, even though it’s rarely explicitly stated as such.
>
> Our assumption of critic convergence, both in its current and previous form, is actually somewhat limited compared with this, as convergence of the agent’s policy network (the implicit assumption underpinning prior work in this area) is generally dependent on (and therefore assumes) convergence of its critic network(s). You’re of course right to note that, in practice, finite and biased data is always what we’re working with. To more accurately characterize the convergence property, at the moment we rely on the convergence property of the base learning algorithm as the worst case scenario since ADOPS will produce an optimality-preserved policy. Our experimental results are promising, and show our method’s ability to effectively improve training in a complex, sparse-reward environment. We’re enthusiastic about expanding these results to additional domains and forms of IM in future work.
>
> Additional responses to your comments/questions, in order:
>
> Thank you for your correction about Equation (3); we’ve edited it to read $F_t$ rather than $F’_t$.
>
> We’ve added an explanation/definition of $\pi_1$ and $\pi_2$ below Equations (13) and (14), on your recommendation. These are simply two (potentially distinct) extrinsically optimal policies.
>
> $\bar{a}$ is an extrinsically suboptimal action for a given state $s$.
>
> These assumptions you mention (being episodic and/or future-agnostic) are not necessary for our method, and can be violated freely without consequence. This is an important strength of ADOPS that you’re right to note we forgot to keep stressing the significance of about halfway through the paper. To fix this, we’ve added an additional note of our ability to relax these assumptions in the conclusion of the paper. Thank you for pointing this out.
>
> Our use of the phrase “preexisting network critics” is admittedly confusing, and we’ve replaced it in the paper for clarity. We meant “preexisting” just in the sense that these networks were already part of the architecture and algorithm being used, and so using them in our method introduced no additional computational overhead. The critic networks update their parameters throughout training, just like in a standard PPO algorithm.
>
> Thank you again for your time and helpful feedback.

---

### Official Review · Reviewer_VPTk · 2025-03-15

**Overall Recommendation:** 4

**Summary:**

The paper focuses on improving performance in complex, exploration-heavy environments with long-duration episodes. In particular, the paper focuses on reward shaping methods that shape rewards while maintaining optimality.

The paper proposes a new reward shaping approach called Action-Dependent Optimality Preserving Shaping (ADOPS). The paper claims that

(a) ADOPS allows for reward shaping that previous potential-based reward shaping methods do not.

(b) ADOPS allows for intrinsic cumulative returns to be dependent on agents’ actions while still preserving optimality.

(c) ADOPS empirically improves performance over baseline intrinsic motivation in complex, extremely sparse environments where preexisting methods for preserving optimality fail.

## update after rebuttal
I am happy with the rebuttal. It addressed my major concerns. I think the paper has sufficient evidence for the claimed contributions.

**Claims And Evidence:**

For claim (a), Theorem B.1 shows that there are are reward shaping functions that previous methods such as GRM cannot use. Moreover, Theorem B.2 shows that the proposed ADOPS conserves the set of optimal policies.

For claim (b), ADOPS is designed to have action dependent reward shaping and the theoretical proof shows that ADOPS still preservers optimality.

For claim (c), the paper compares ADOPS and baselines in the Atari game Monte Zuma's Revenge. In this case, using a single benchmark is in my opinion sufficient and a good choice. The results are thoroughly analyzed and explained. Nevertheless, saying that the approach improves performance in more than one experiment (the word "environments", plural, is used) is a slight over-claim w.r.t. the empirical evidence and the description should be modified.

**Essential References Not Discussed:**

To me the references look fine. The paper has a well defined scope.

**Experimental Designs Or Analyses:**

The experimental design makes sense. The experimental analysis is of high quality motivating the choice of hyperparameters and other design choices well.

**Methods And Evaluation Criteria:**

The paper compares ADOPS and baselines in the Atari game Monte Zuma's Revenge. In this case, this makes sense. The paper contains theoretical proofs and the chosen benchmark and the resulting analysis provide detailed knowledge of the methods.

**Other Comments Or Suggestions:**

In "While it would be ideal, it is usually not feasible to implement Equation (41), as it requires an accurate estimate of the optimal value function.", should (41) be (15)?

In multiple locations, the paper uses phrases of the form "We will begin". These can be rephrased to "We begin". The word "will" is not necessary.

Figure 1 shows result plots w.r.t. training steps. What does training steps mean here? How many time steps is one training step?

**Other Strengths And Weaknesses:**

The paper is well written.

**Questions For Authors:**

I did not fully understand the text that says why Equation (13) leads to "the first of these conditions says that every action that would be optimal without IM must remain optimal after the addition of the IM". Can you please explain this in more detail?

**Relation To Broader Scientific Literature:**

The paper provides a novel contribution in the specific domain of reward shaping. The proposed approach shows how to do reward shaping in a more complex action dependent way while maintaining optimality w.r.t. the original reward function.

**Theoretical Claims:**

I did not notice problems with the proofs. However, I did not check the proofs in detail.

---

> ### Author Rebuttal · Authors · 2025-04-01
>
> Thank you for taking the time and effort to review our work. We appreciate your positive feedback.
>
> We fixed the reference to Equation (41) to Equation (15). Thank you for noticing this error.
>
> We removed unnecessary use of future tense as you suggested.
>
> Each "training step’’ is a rollout and subsequent set of gradient updates that occurs every 128 time steps across each of 32 workers. We use the same architecture and training procedure as in the original RND paper (Burda 2019). We’ve updated our appendix to define "training step’’ more clearly.
>
> We’ll explain a bit more what we meant when we said Equation 13 entails that "every action that would be optimal without IM must remain optimal after the addition of the IM.’’ There are multiple extrinsically optimal policies, two of which we can call $\pi_1$ and $\pi_2$. These policies, being extrinsically optimal, have equal $\textit{extrinsic}$ value functions $V_E^{\pi_1} = V_E^{\pi_2}$, but that doesn’t necessarily mean $\textit{ a priori}$ that they have equal intrinsic value functions. If they happen to not have equal intrinsic value functions, then one will be greater than the other, and thus the extrinsically optimal policy with the lower intrinsic value function will no longer be optimal (because there exists another policy that would obtain greater expected value). In order to ensure that this doesn’t happen, and that all extrinsically optimal policies are also optimal intrinsically, we need to assert that their intrinsic value functions (and thus their combined intrinsic+extrinsic value functions) are equal to each other. Admittedly, we had worded this somewhat confusingly, as we reference "actions,’’ but Equation (13) deals explicitly only with policies. We’ve amended our explanation to say  "every policy that would be optimal without IM must remain optimal after the addition of the IM.’’ Thank you for bringing this to our attention.
>
> Thank you again for your time, effort, and kind words.

---

### Official Review · Reviewer_4wwG · 2025-03-19

**Overall Recommendation:** 3

**Summary:**

This work introduces Action-Dependent Optimality Preserving Shaping (ADOPS), a general framework for modifying intrinsic rewards to a form that preserves the set of optimal policies. The proposed framework is general enough to capture not only potential-based reward shaping (PBRS) functions, but also reward shaping functions that cannot be written in potential-based form. Indeed, contrary to PBRS, ADOPS allows for reward shaping functions with the property that the intrinsic cumulative returns depend on agent's actions while still preserving the set of optimal policies. The authors evaluate their framework in the Montezuma's complex Atari Learning Environment, which is characterized by very sparse rewards, and show that ADOPS outperforms all other baselines, some of which struggle to make any progress at all.

## update after rebuttal
I have read the rebuttal carefully and I will keep my score unchanged. The authors have mostly addressed my questions and comments. Something to point out is that the revised proof is based on the concept of a stable policy, and I am not so clear how common such proof strategies are in the RL Literature. Can for instance the authors make any connections (if applicable) to the prior literature? Furthermore, I still feel the proof is not so easy to follow, even though the revised proof is better written.

**Claims And Evidence:**

Overall, I feel that the paper does not make unsubstantiated claims, in the sense that it provides both theoretical derivations as well as empirical evidence for the various claims it makes. That said, I feel that the empirical evidence is limited, as I explain in the next sections. I also have some concerns regarding the theory, which I discuss below.

**Essential References Not Discussed:**

My main concern regards the literature on Intrinsic Motivation. The authors cite some important approaches, namely, count-based exploration (Bellemare et al., 2016), Intrinsic Curiosity Module (ICM) (Pathak et al., 2017), and Random Network Distillation (RND) (Burda et a l., 2018a; 2019). All these papers are important but not recent. I think the authors could have included some papers from the more recent literature.

This could also be helpful in relation to the baselines in the experimental section. The authors use RND only from the IM literature, which is rather old. It would be great if they could show that their approach outperforms more recent approaches from the IM literature as well. This could broaden the scope of the work - do the authors want to show that ADOPS just outperforms PBRS approaches, or that it can in fact be competitive w.r.t. recent IM approaches (see also my earlier comment)?

**Experimental Designs Or Analyses:**

I did not have any particular concern with the design. The authors made changed some things from the prior literature, like not clipping external rewards, but everything is explained sufficiently.

The experimental analysis is quite deep, as the authors dive deeper into all methods.

**Methods And Evaluation Criteria:**

The fact that the authors only test their approach against a single environment is a bit problematic. The Montezuma's complex Atari Learning Environment is indeed a challenging environment with very sparse rewards, but results with more environments would have made the various claims more convincing.

- As an example, Random Network Distillation was testing on multiple environments such as Gravitar, Montezuma’s Revenge, Pitfall, PrivateEye, Solaris, and Venture. The recent works by Forbes (cited in the paper) also test on several environments, namely, MiniGrid DoorKey, Cliff Walking, and Longer Cliff Walking. Not all environments are equally interesting, and obviously the authors would want to choose the challenging ones with sparser rewards. My point is simply that one environment does not provide significant evidence.

As far as the baselines are concerned, I like the fact that the authors have decided to experiment with three of the most recent
potential-based approaches: PIES, PBIM, and GRM. That said, it is also important to understand how the proposed framework compares against more traditional approaches based on curiosity and intrinsic rewards (even if these do not have theoretical guarantees). Random network distillation is an excellent candidate, but rather old. There are more recent works  like ""Never give up :Learning directed exploration strategies" by  Badia et al. from ICLR 2020. It would have been interesting to see how the proposed framework would compare against the state of the art in intrinsic (but not PBRS) rewards.

**Other Comments Or Suggestions:**

- It would be nice if the authors could improve the exposition of the proof of Theorem 5.3. It might be helpful to structure the proof in a case-based format, e.g., Case 1: ...., Case 2: ...., etc. Currently, there is a lot of text, but I feel that more rigorous symbols and derivations might improve the exposition.
- It would be nice if the authors could experiment with more environments and/or recent sate-of-the-art baselines from the IM literature.

**Other Strengths And Weaknesses:**

Strengths
- The framework is a nice contribution that goes beyond the current PBRS literature. Action-dependent but policy preserving reward shaping is a novel and promising idea.
- The theoretical derivations are generally done with care.
- The results on the Montezuma's complex Atari Learning Environment are good, and demonstrate superiority w.r.t. recent PBRS approaches and even RND.
- The experimental analysis is interesting.

Weaknesses
- Proof 5.3 is hard to read with confusing notation and some non-rigorous statements.
- Using a single environment is not as convincing as experimenting with several environments.
- RND is not a recent approach from the IM literature. It would be interesting to know how this framework compares against more recent state-of-the-art IM frameworks.

**Questions For Authors:**

Please address the various concerns.

**Relation To Broader Scientific Literature:**

The paper pushes the envelope in the reward-shaping literature by proposing the novel Action-Dependent Optimality Preserving Shaping (ADOPS). This framework can unify PBRS approaches; in addition to that, it can incorporate shaping reward that fall outside the PBRS paradigm. I find the framework quite powerful and interesting.

**Theoretical Claims:**

Overall, I feel that the authors have done a good job with the derivations. Some concerns:
- In (38), I think $F'_t$ should read $F'_j$.
- In (39), the notation $U_t^{I_{old}}$ locks strange. Why "old"?
- The proof of Theorem 5.3 may be correct, but it is not easy to read. One confusing thing concerns the policy. The authors claim that $\pi$ is the current policy followed during training. I assume this policy refers to the ADOPS reward, not the original reward. So, essentially the current policy outputs the action it thinks is best in terms of the shaped ADOPS reward. The notation with the Q- and V-functions is also quite confusing, because there is an implicit dependence on the timestep and the policy.
- Another issue with the proof of Theorem 5.3 concerns Assumption 5.2, which guarantees adequate exploration. I understand the authors need that assumption in the proof of the theorem. But this is still very high-level, and not completely rigorous. It is more of the big idea. I am not sure whether it can count as a formal argument though, but I could be wrong.
- In (35), (36) and even other equations, the authors do not sum over all possible states $s'$ The Bellman operator should contain the summation over all possible next states though, weighted by the transition probabilities.
- I was not able to figure out what (29)-(31) show. In particular, how do we go from (29) to (30)?

---

> ### Author Rebuttal · Authors · 2025-04-01
>
> Thank you for your thorough feedback, and for the time and energy it took to read our paper in such an in-depth manner. We appreciate your insightful comments, and our resulting edits to the paper have improved it substantially.
>
> We acknowledge that further work on a wider suite of environments than Montezuma’s Revenge is merited, and are excited to explore this territory in future work: we are currently working to extend our empirical evaluations to other Atari environments for a follow-up work. We limited ourselves to one environment in this work due to time and compute constraints, and chose Montezuma’s Revenge due to its status as a well-tested benchmark known for being extremely sparse and difficult to learn. We’d also like to emphasize, as you’ve pointed out, that ours is a plug-and-play method with theoretical guarantees, and this represents the most complex environment in which such a method has been tested. Additionally, as you note, it would be interesting to apply our methods to shaping rewards other than RND, particularly more recent, SOTA methods such as Never Give Up. We’re excited to do so in future work; we chose RND as a base IM method for its relative simplicity and ubiquity, as a good baseline for demonstrating the initial efficacy of our method. We limited ourselves to one IM method due, again, to ever-present time and compute constraints (particularly here, wherein our experimental setup requires testing each IM method with each plug-and-play method, and so an additional one would almost double our required computational budget). We consider our experimental results in Montezuma’s Revenge to be strong contributions for an initial work, particularly when paired with our theoretical contributions.
>
> We will next address your comments in the “Theoretical Claims” section in the order that they appear.
>
> We’ve corrected Equation 38 as appropriate. Thank you for catching this.
>
> We agree that  $U^{I_{old}} t $ is a somewhat odd notation, and furthermore unnecessary to condense what it is meant to represent at all. We’ve eliminated it entirely, and replaced it with the full text of what it notates, namely $\sum_{j=t}^{N-1} \gamma^{j-t}F’_j$ (the return of the “old” intrinsic reward, unmodified by GRM).
>
> We have significantly updated the text of our proof of Theorem 5.3 for clarity and rigor. We’ve also added a preamble to the full text of it explaining the outlines of the proof. We further explain our modifications to the text of the proof, including more formally-defined assumptions, in our response to Reviewer cdE4. These changes include moving the $P(s’|a,s)$ term to the distribution of an expectation function, thus avoiding the need to include any (initially omitted, as you noted) summation in Equations (35) and (36), among others.
>
> A fully updated version of Section 5.2 is included in the below link, to peruse the changes to this proof at your leisure.
>
> https://drive.google.com/file/d/1XTZ3VSuIDFaqHM_NxsjBpQEhkR6DFOyM/view?usp=sharing
>
> The transition from Equation (29) to (30) is essentially the same as the transition from (24) to (28), except that all the $C_1, C_2, C_3 = 0$, and the $V_E + V_I$ terms do not drop out, as they’re inside a max function rather than an argmax. This was admittedly confusing, and also unnecessary to the flow of the proof. We’ve omitted these equations in the new version, in favor of a more rigorous explanation.
>
> As for addressing more literature, we’ve added acknowledgements of more recent approaches to Section 2, including Never Give Up and  AIRS (suggested by another reviewer).
>
> Thank you again for the review, particularly the kind words about the novelty of our framework and theory; you clearly read the paper very closely and understand what we’re trying to do, and it’s nice to feel “seen.”

---

### Decision · Program_Chairs · 2025-05-01

**Decision:**

Accept (poster)

**Comment:**

This submission has received four expert reviews with all reviewers acknowledging the authors' responses. There is a consensus among the reviewers that the proposed reward-shaping method (ADOPS) unifies and extends previous approaches in the reward shaping literature. The reviewers found the paper generally well-written and the evidence that supports the claims in the paper convincing. Importantly, several of the reviewers' concerns/misunderstandings were addressed/clarified during the rebuttal. Among the remaining concerns, the most important is the limited experimental evaluation in different environments. This is a rather significant concern which in this case is countered by the superior performance of the proposed method against a wide range of baseline reward shaping methods in the particularly thorny (due to the scarcity of rewards) Montezuma's revenge environment. While this is enough to recommend acceptance, it limits the potential of the paper for a stronger recommendation.

Going forward, I strongly encourage the authors to include the insights/limitations and key points from the discussion with the reviewers (the updates in the theoretical assumptions and theoretical proofs are much appreciated) in the final version of the paper as this will considerably alleviate similar concerns for the paper's readership.